# Review: Proteomic Techniques for the Development of Flood-Tolerant Soybean

**DOI:** 10.3390/ijms21207497

**Published:** 2020-10-12

**Authors:** Xin Wang, Setsuko Komatsu

**Affiliations:** 1College of Agronomy and Biotechnology, China Agricultural University, Beijing 100193, China; 2Faculty of Environmental and Information Sciences, Fukui University of Technology, Fukui 910-8505, Japan

**Keywords:** proteomics, omics, soybean, flooding, stress tolerant, stress response

## Abstract

Soybean, which is rich in protein and oil as well as phytochemicals, is cultivated in several climatic zones. However, its growth is markedly decreased by flooding stress, which is caused by climate change. Proteomic techniques were used for understanding the flood-response and -tolerant mechanisms in soybean. Subcellular proteomics has potential to elucidate localized cellular responses and investigate communications among subcellular components during plant growth and under stress stimuli. Furthermore, post-translational modifications play important roles in stress response and tolerance to flooding stress. Although many flood-response mechanisms have been reported, flood-tolerant mechanisms have not been fully clarified for soybean because of limitations in germplasm with flooding tolerance. This review provides an update on current biochemical and molecular networks involved in soybean tolerance against flooding stress, as well as recent developments in the area of functional genomics in terms of developing flood-tolerant soybeans. This work will expedite marker-assisted genetic enhancement studies in crops for developing high-yielding stress-tolerant lines or varieties under abiotic stress.

## 1. Introduction

Climate change, which occurs naturally in addition to human activities, is a major global concern [1]. Climate change produces alterations in rainfall intensity and patterns, and extremes in precipitation are more frequently limiting the production of food products [2]. The majority of arable lands in the world are prone to unfavorable environmental conditions [3]. Out of all of them, flooding stress causes growth inhibition and ultimately death in most crop species by limiting energy production [4]. Due to the scarcity of available oxygen in waterlogged soil, flooding stress leads to partitioning of oxidative systems in plants, resulting in the accumulation of ethylene and carbon dioxide [5], which impairs and damages root extension. These findings indicate that plant growth is severely affected by flooding due to the limited oxygen levels.

Rice is one of the few flood-tolerant crops; however, it cannot grow under extended periods of complete submergence. Most lowland rice genotypes generally adopt an escape strategy, which is characterized by ethylene-mediated rapid elongation promoted by gibberellins associated with carbohydrate consumption [6]. In deep-water rice, the escape strategy is regulated by two ethylene-responsive factors, SNORKEL1 and SNORKEL2, that trigger considerable internode elongation via gibberellins during flooding [7,8]. Some varieties of rice tolerate complete submergence by inhibiting the elongation of submerged stems and leaves, which suppresses the consumption of carbohydrates instead [9]. The gene related to such submergence tolerance is SUBMERGENCE1, which encodes a variable cluster of ethylene-responsive factor genes [10,11]. In such submergence-tolerant rice, elongation of stems and leaves was inhibited by suppression of the increase in ethylene concentration, which also decreases plant sensitivity to gibberellins. The plant maintains a balance between carbohydrate supply and demand.

Soybean is considered a unique leguminous crop, because its seed is a rich source of protein, essential amino acids, oil, and metabolizable energy [12]. Although soybean is one of the major agricultural crops, it is particularly sensitive to flooding stress [13]. Its plant growth and grain yield are markedly reduced in flooded soil [13]. When soybean was exposed to flooding at the vegetative growth stage or reproductive growth stage, grain yield and quality were reduced [14]. In addition, secondary aerenchyma was formed and worked as an oxygen pathway under flooded conditions [15]. Furthermore, flooding stress impaired plant growth by inhibiting root elongation and reducing hypocotyl pigmentation [4]. In the root of soybean, ubiquitin-mediated proteolysis was activated [16] and cell death was detected under flooding [17]. These results indicated that flooding causes damage to soybean at the early stage of growth.

To understand flood-response mechanisms in soybean, proteomic analyses were performed. Under flooding, soybean seedlings showed differential regulation of proteins involved in signal transduction, hormonal signaling, transcriptional control, glucose degradation/sucrose accumulation, alcohol fermentation, gamma-aminobutyric acid (GABA) shunt, suppression of reactive oxygen species (ROS) scavenging, mitochondrial impairment, ubiquitin/proteasome-mediated proteolysis, and cell-wall loosening [18,19,20]. Although flood-response mechanisms of soybean have been explored using proteomic techniques, flood-tolerant mechanisms have not yet been uncovered. For this reason, soybean materials with flooding tolerance have been generated [21,22,23]. In this review, flood-response mechanisms are summarized with proteomic data with subcellular and post-translational modifications (PTMs). In addition, soybean survival under flooding, which is defined as a tolerance mechanism, is discussed with results of comprehensive analyses in soybean materials with flooding tolerance. Finally, interactions of the molecular mapping of omics data are proposed.

## 2. Morphological and Physiological Effect of Flooding Stress on Soybean

There is no doubt that flooding stress alters soybean morphology at the vegetative and reproductive stages [13]. Morphological traits, including seedling growth, hypocotyl pigmentation, plant flowering, seed filling, and aerenchyma formation, are influenced by flooding stress, leading to retarded plant growth and reduced seed yield. Along with morphological alterations, physiological events are triggered by flooding. The morphological and physiological effects of flooding on soybean are summarized in Appendix A.

A plethora of findings present adverse effects of flooding on soybean growth and development within short- and long-term stress (Appendix A). Overall, flooding stress suppressed elongation of radicle/root/hypocotyl and reduced root number/root surface/leaf area, leading to inhibition of plant growth [24,25,26,27,28,29,30,31,32,33]. However, the effect of flooding on fresh weight of plant and adventitious root was dependent on prolonged duration of stress and growth stage of soybean. For example, the fresh weight of soybean increased in 3 h-flooded soybean, while it decreased with prolonged stress [29,33]. The length of adventitious root was shorter in 5-day-old soybean exposed to flooding for 3 days than untreated seedlings [34], while increased adventitious root was observed in 28-day-old soybean subjected to flooding for 21 days [24]. Furthermore, flooding slightly increased days to maturity [13] and caused severe yield loss of soybean regardless of growth stage [25,34]. In addition, formation of secondary aerenchyma was promoted by stress, which was considered as morphological acclimation of flooding stress [15,35,36]. These observed morphological alterations indicate the injurious effects of flooding on the agronomic traits of soybean; however, promising traits such as adventitious root and aerenchyma formation might enhance soybean flooding tolerance.

A series of cellular processes have been illustrated in soybean, including cell wall reorganization, calcium signaling, cellular metabolism, and hormone signaling (Appendix A). Suppressed cell-wall lignification [37,38], cell-wall synthesis [17], as well as flooding-induced plasma membrane [39] contributed to cell-wall reorganization in soybean root. It was shown that calcium ion in the root tip and cotyledon was induced by flooding, and calcium-related proteins such as calreticulin and pyruvate decarboxylase, which were associated with protein folding and pyruvate catabolism, responded to flooding [40,41,42,43]. The deleterious effect of flooding was evident with cell death, which occurred in 2-day-old soybean exposed to flooding for 1 day and as long as 6 days [17,44,45]. Elucidation of cellular metabolism showed that protein metabolism [16,38,42,46,47], RNA metabolism [38,48], energy management [44,49], carbohydrate metabolism [50,51], and carbon-nitrogen flux [51,52,53] responded to flooding. In addition, hormone signaling coordinated with flooding response, showing that ethylene and abscisic acid (ABA) acted with protein metabolism and energy conservation [33,54,55,56]. It was noticeable that both calcium signaling and ABA were induced by flooding and triggered protein metabolism and energy metabolism, and protein phosphorylation was involved in the integration of calcium–ABA signaling. Therefore, a proteomic approach is beneficial to identify master regulators involved in stress signaling in soybean under flooding.

## 3. Plant Omic Analysis to Understand Flood-Response Mechanisms in Soybean

Plant omics is becoming important in the scientific community because of the urgent need to address issues facing humanity with regard to agricultural production, environmental decontamination, and ecological sustainability [57]. Proteomics has been utilized in the plant field since the beginning of the twenty century and it is still an irreplaceable approach to explore protein profiles in response to environmental stimuli [58]. As flooding sensitive legume with economic relevance in food production, progressive findings of flood-response mechanisms in soybean were achieved through proteomic analysis. Plant omics including proteomics, transcriptomics, metabolomics, as well as bioinformatics is reviewed in Appendix A.

### 3.1. Proteomics to Understand Flood-Response Mechanisms in Soybean

Proteomics with crude proteins helps us to obtain proteins produced and utilized by organisms to sustain various processes required for flooding responses. To date, a series of protein profiles in flooded soybean has been acquired (Appendix A). In 2- and 3-day-old soybean, protein profiles altered within a few hours of flooding, and alterations of proteins involved in calcium signaling and carbohydrate metabolism were recognized as flooding response [41,50]. When flooding is prolonged, cell death in root occurred and suppressed cell-wall synthesis/protein metabolism was implicated as a cause of stress injury [17,28,31]. Besides this, under flooding, alterations of proteins in the V2-growth stage were similar to germinating seeds, with carbohydrate consumption and programmed cell death [59]. Furthermore, it was indicated that protein synthesis, cell-wall metabolism, energy consumption, and antioxidant metabolism were conserved by the flooding response, which was regulated by a core group of proteins in soybean seedlings [17,29,50,60]. In addition, proteins involved in cell reorganization, secondary metabolism, and glycolysis protected flooded soybean from cell death, energy crisis, and toxic radicals, thereby promoting post-flooding recovery [30,61,62].

Based on flooding responsive maps sketched by crude proteins, subcellular proteomics has been applied to interpret events in targeted subcellular and bridge interactions within cell compartments. The progress of the flooding response acquired from subcellular proteomics in soybean is presented (Appendix A). Under flooding, cell-wall metabolism was highlighted by proteomic analysis using crude proteins, and biosynthesis of jasmonic acid and ROS scavenging contributed to suppressed lignification in root including hypocotyl [37]. In the plasma membrane, proteins related to the antioxidative system, stress, and signaling played roles in protecting the cell from oxidative damage, protein degradation, and ion homeostasis under flooding [39]. Compared with cell wall and plasma membrane, more studies have focused on nuclear aspects and addressed the suppressed RNA metabolism and activated ABA signaling in response to flooding with exposure from 3 to 48 h [37,48,54,55], whereas DNA repair via acceleration of poly-ADP-ribosylation and signaling transduction via interaction of RACK1 with 14-3-3 protein coped with prolonged flooding [63,64]. Energy consumption was evidenced with impaired electron-transport chains in mitochondria and proteins involved in the tricarboxylic acid (TCA) cycle were determined by mitochondrial proteomics in response to flooding [65,66]. In addition, endoplasmic reticulum proteomics implied that flooding suppressed protein synthesis and caused dysfunction of protein folding, leading to the reduction of glycoproteins [42,46]. Although these studies highlighted stress responses in specific subcellular regions, receptors for flooding have not been identified and stress signals within cells need further investigation.

PTMs regulate protein activity, localization, as well as protein–protein interactions in cellular processes, leading to elaborate regulation of plant response to environmental stimuli [67]. In soybean, PTM-mediated flooding response is illuminated with developed proteomics (Appendix A). At initial flooding, hormone regulation of ethylene and ABA participated in flooding tolerance via phosphorylation of eukaryotic translation initiation 4G, zinc finger/BTB domain-containing protein 47, glycine-rich protein, and rRNA processing protein Rrp5 [33,55]. Flooding altered phosphorylation status through dephosphorylated proteins involved in protein folding/cell structure and phosphorylated proteins involved in energy production [50,68]. Besides this, glycosylation and ubiquitination play roles in protein synthesis and degradation under flooding. For example, a 2-day flood inhibited the biosynthesis of glycoproteins [42], while it activated ubiquitin-mediated protein degradation [16]. Under flooding, *S*-nitrosylation proteins were enriched in glycolysis and fermentation, and *S*-nitrosylation modulated the stress response via activated or deactivated forms of sugar-degrading enzymes [69]. Collectively, these findings indicate that protein phosphorylation is an initial response to flooding and it modulates protein synthesis and RNA processing; however, glycosylation, ubiquitination, and *S*-nitrosylation modulate protein metabolism, carbohydrate catabolism, and energy production in soybean with extended stress duration.

### 3.2. Plant Omics of Transcriptomics, Metabolomics, and Bioinformatics

Plants respond to stressful conditions through changes in omic profiles and enormous progress has been achieved in the area of plant omics [18]. Current findings of transcriptomics, metabolomics as well as bioinformatics collected from flooded soybean are summarized in Appendix A. More than 6000 genes were identified in the root and hypocotyl of soybean exposed to flooding for 6 and 12 h, and downregulated genes of pyrophosphate dependent on phosphofructokinase and upregulated genes encoding small proteins played roles in flooding acclimation [70]. In the leaf, 3498 differentially expressed genes were identified, and genes involved in cell-wall precursors and starch content served as adaptive mechanisms for soybean survival from flooding [71]. Tamang et al. [72] reported that although the expression of organ-specific genes was demanded in different organs of soybean exposed to submergence, conserved responses were invoked in shoot and root. These documents indicate that carbohydrate catabolism might play a conserved role in flooded soybean regardless of plant organs. Furthermore, flooding disturbed the balance of the carbon–nitrogen ratio in soybean root, which was evident with accumulated GABA and increased metabolites involved in the TCA cycle [52,65]. Although carbohydrate catabolism and carbon–nitrogen influx were pointed out by transcriptomic and metabolomic studies of flooded soybean, reference maps of protein profiles were sketched using proteomics data stored in the Soybean Proteome Database [73,74,75]. The database was not a simple tank for data storage; however, multiple ome data were acquired from uniform experimental conditions. Moreover, the Soybean Proteome Database provided temporal profiles of omes, which were characterized with unified temporal-profile tags. Although proteomic data were collected from gel-based proteomic studies at the beginning, the database gradually updated with the information of gel-free proteomics, subcellular proteomics, PTMs profiles, protein categorization, and protein–protein interactions from different organs in flooded soybean. Therefore, the Soybean Proteome Database has provided comparative and integrated omes in soybean, assisting in the illustration of flood-response and -tolerance mechanisms in soybean.

### 3.3. The State of Proteomic Analysis of Other Crops under Flooding Stress

A considerable volume of proteomic studies of flooding response focuses on soybean, while works on other crops such as rice [76,77,78], maize [79,80,81,82], wheat [83,84,85,86], rapeseed [87], barley [88], and alfalfa [89] are collected in Table 1, which assists in addressing the commonality of flooding response in crops. ROS serves as an intermediator to percept environments during seed imbibition and ROS levels are maintained at a level which triggers cellular events associated with germination [90]. During seed germination, proteins related to redox were enriched, with decreased abundance in anoxic rice coleoptile [76] and elevation of hydrogen peroxide was a component of the pathway that induced alcohol dehydrogenase expression under low oxygen [91], indicating that ROS acted as a signaling molecule in alcohol fermentation. It was noticeable that phenylpropanoid biosynthesis and fatty acid metabolism were ubiquitous stress responses to drought, salinity, and submergence during seed germination in wheat; however, starch and sucrose metabolism was indicated as a submergence-specific response for germination [86]. Whether crops could prepare for the second round of flooding stress was examined and it was showed that waterlogging priming alleviated the yield loss of wheat through ethylene signaling, which improved leaf photosynthesis by promoting stomata opening under waterlogging [84]. Similar to the flooding response in soybean, cellular events such as electron transfer chain, programmed cell death, energy metabolism, ethylene production, and cell-wall hydrolysis were induced by flooding in other crops, indicating that these metabolisms were conserved by flooding responses within crop species. Currently, subcellular proteomics, PTMs, as well as post-flooding responses are rarely considered in other crops, and thus proteomic studies in soybean could be used to unveil molecular mechanisms underlying the core metabolism in different crops towards flooding stress.

## 4. Proteomic Analysis to Understand Flood-Tolerant Mechanisms in Soybean

Narrow genetic background and limited genetic diversity restrict the development of new elite soybean cultivars; however, man-made mutations produce numerous genetic variations which could be used to select lines with ideal plant architecture and tolerant characteristics [92]. Besides traditional germplasm screening, several approaches are adopted to promote soybean growth under flooding, such as transgene technology, millimeter-wave irradiation, and chemical supplication of phytohormone, plant-derived smoke, and nanoparticles (NPs). As a cutting-edge molecular technique, the application of proteomics on flood-tolerant crops was reviewed, reinforcing its potential roles to assist in developing high-yield flood-tolerant materials under flooding [22]. Elucidations of proteomics on flood-tolerant mechanisms in soybean are outlined in Table 2.

### 4.1. Proteomics Using Generated Flood-Tolerant Lines/Varieties

#### 4.1.1. Soybean Varieties with Flooding Tolerance

Bottlenecks in genetic diversity challenge germplasm screening for flood-tolerant lines of soybean, while comparative proteomics assists in the elucidation of flooding tolerance in soybean. Nanjo et al. [31] examined the flood-tolerant index of 128 soybean varieties, including survival rate of plant, lateral root development, radicle elongation, and symptoms of flooding injury; these varieties were classified into three categories based on the index. Proteomic study of the soybean radicle collected from these categories showed that RNA binding/processing-related proteins and flooding stress indicator proteins correlated with flood-tolerant index [31]. Lin et al. [105] reported that Qihuang 34 was a flood-tolerant cultivar, which was indicated with the highest survival ratio of seedling, and metabolic pathways relevant to carbon metabolism, mitogen-activated protein kinase signaling, fatty acid degradation, as well as isoflavonoid biosynthesis were enriched via differently changed genes and proteins under flooding. In addition, it was shown that glycolysis was activated for ATP production for plant survival; however, decreased lignin caused plant softening under submergence for a long period of time [105]. These findings are consistent with master regulations of RNA metabolism and carbohydrate supply in flooded soybean that were discussed above. In addition, this raises the possibility that secondary metabolites might improve flooding tolerance as for isoflavonoid.

#### 4.1.2. Mutant Soybean with Flooding Tolerance

Deepwater- and submergence-tolerant rice utilize the strategy of internode elongation or remain stunted to survive flooding stress [8]. Zaman et al. [106] found contrasting tolerant mechanisms in pea seed at germination stage under waterlogging, including a quiescence strategy where the seed reserved the energy preserved and used an escape strategy involving rapid germination utilizing energy from protein/lipid metabolism. To characterize flood-tolerant mechanisms in soybean, a flood-tolerant line was generated by physical mutagenesis of gamma-ray irradiation with six-time tolerant screening [44]. Proteomic analysis using the flood-tolerant mutant indicated that proteins involved in cell-wall loosening did not increase in the mutant compared with wild-type soybean, thereby preserving the viability of the root tip and permitting its rapid growth during the post-flooding stages [44]. Due to the oxygen-depleted environment arising from flooding, a crisis in ATP availability occurred [107]; however, to manage the energy crisis, the balance of glycolysis aided in soybean survival from submergence [38,105] and enhanced fermentation was necessary for the acquisition of flooding tolerance [44]. These findings indicate that activated fermentation and glycolysis confer soybean flooding tolerance to ensure survival. In addition, carbohydrate catabolism might be helpful to build blocks for cell-wall synthesis in a flood-tolerant mutant.

#### 4.1.3. Transgenic Soybean Overexpressed Flood-Response Gene

Biotechnology of genome editing is an ideal approach for precision crop breeding; however, there are no reports of its application to the improvement of flooding tolerance in crops. Transgenic approaches have been widely used to address flood-tolerant mechanisms in *Arabidopsis* [108,109], rice [110], maize [111], and soybean [93]. Under flooding, soybean with heteroexpression of *AtXTH31* presented higher germination rates and longer length of seedling than the wild type, indicating that *AtXTH31* conferred soybean flooding tolerance [112]. Moreover, transcript levels of the *XTH* gene family in soybean were investigated and 23 *GmXTH* were significantly regulated by ethylene in soybean root, indicating that ethylene played a role in GmXTH-mediated cell-wall remodeling under flooding [112]. *GmAdh2* was identified as a root-specific gene in response to flooding, and thinner hypocotyl as well as smaller area of primary leaf were observed in wild-type soybean compared with *GmAdh2*-overexpressed lines, indicating that *GmAdh2*-mediated fermentation contributed to the acquisition of flooding tolerance [93,113]. This limited volume of studies indicates that remodeling root architecture and enhanced anaerobic respiration are feasible to improve soybean tolerance to flooding stress.

#### 4.1.4. Soybean Irradiated with Millimeter Wave

Millimeter waves are typically defined to lie within a frequency of 30–300 GHz and millimeter-wave radiation is an environmentally friendly technology which has been proposed as a useful method to increase the quality characteristics and harvest for agricultural plants [114,115]. Betskii et al. [115] found the main results of millimeter-wave irradiation on plants with considerably increased energy of germination, seed germination rates, and short periods of phenophases. Zhong et al. [94] examined the action of irradiated-millimeter wave on soybean, indicating that irradiation elongated the hypocotyl of soybean without and with flooding. It was reported that soybean seeds pretreated with millimeter-wave irradiation presented promoted growth under oxidative stress arising from flooding through the regulation of activated sugar metabolism and enhanced ascorbate/glutathione metabolism [94]. In addition, radiation altered the intracellular structures of lipoproteins, mitochondria, and microsomes [115]; however, flooding caused lipid peroxidation and mitochondrial impairment in soybean [65,72]. Collectively, these results indicate that seed priming with millimeter-wave irradiation is feasible to improve soybean performance under flooding, which might modulate redox scavenging in plant cells, especially in mitochondria.

### 4.2. Application of Chemicals for Flooding Tolerance

#### 4.2.1. Abscisic Acid Treatment

Flooding causes the rapid accumulation of ethylene in plant cells, and ethylene-mediated response assists in flooding acclimation through ethylene response factor, ROS, sugar, and nitric oxide [116]. It was reported that accumulated ethylene stimulated petiole elongation through inhibiting the biosynthesis of ABA, indicating contrasting interactions between ethylene and ABA in flood-tolerant *Rumex palustris* [117]. However, Hwang and Vantoai [118] found that ABA enhanced the anoxic survivability of maize seedling to 87% through the activated synthesis of new proteins and increased enzyme activity of alcohol dehydrogenase. Under flooding, the content of endogenous ABA was lower in waterlogging-tolerant soybean than susceptible lines, indicating better development of aerenchyma cells in tolerant lines than in sensitive plants [119]. The contribution of exogenous ABA to soybean flooding tolerance was investigated by proteomics. It was shown that proteins involved in glycolysis, vesicle transport, and cell organization decreased under flooding in the presence of ABA; however, endogenous ABA enhanced flooding tolerance through the control of energy conservation via the glycolytic system [54]. These results indicate that although ABA treatment might suppress the formation of aerenchyma cells, energy conservation by controlling sugar metabolism is the major strategy utilized by soybean for survival from flooding stress.

#### 4.2.2. Plant-Derived Smoke Treatment

Smoke derived from burning-plant materials and smoke treatment improved seed germination and plant tolerance to abiotic stresses [120,121,122]. In soybean, smoke treatment increased the length of flooded seedlings and recovered the inhibited effect of flooding after water removal [95,96]. Proteomic studies showed that the metabolisms of energy production and ROS scavenging contributed to tolerance towards flooding, while processes of carbohydrate metabolism and cell-wall synthesis assisted in plant growth during the post-recovery stage [95,97]. Zhong et al. [96] found that smoke treatment improved soybean growth through the regulation of nitrogen–carbon transformation via ornithine synthesis and protein degradation via ubiquitin proteolysis under flooding. Study of maize radicle at the onset of smoke treatment showed that smoke responsive genes between stress- and ABA-related genes were overrepresented, indicating that smoke might play similar roles to ABA in plant acclimation to unfavorable environments [120]. Furthermore, the structure of karrikins was similar to the D-ring of strigolactones and they shared signaling pathways during plant development, which acted in parallel to regulate MAX2 activity in a ligand-dependent manner [123,124]. Taken together, these findings indicate that smoke treatment enables soybean tolerance against flooding stress; however, the involvement of smoke-induced signal transduction and the interaction between ABA and strigolactones in response to flooding need clarification.

#### 4.2.3. Nanoparticle Treatment

NPs are typically ultrafine particles with sizes of less than 100 nm with increasing application in the agricultural sector due to rapid developments in nanotechnology [99,125]. Previously, NPs were considered pollutants with toxicity for soybean growth [98]; however, increasing evidence indicates the positive roles of NPs on flooded soybean [100,101,102,103,104]. Al_2_O_3_-NPs promoted the growth of flooded soybean through controlling energy metabolism and cell death [100]; however, Ag-NPs shifted the metabolism from the fermentative pathway towards normal cellular processes under hypoxia conditions [102,103]. Yasmeen et al. [101] reported that Al_2_O_3_-NPs increased the survival percentage of soybean recovered from flooding through the regulation of *S*-adenosyl-l-methionine-dependent methyltransferases and enolase, indicating that modification of methylation and glycolysis aided in soybean recovery from prolonged stress. Actions of NPs on plant growth are largely dependent on size, concentration, and stability. For example, soybean growth was improved by biosynthesized Ag-NPs compared with chemically synthesized Ag-NPs [104]. With regard to flooding-mediated energy metabolism, Al_2_O_3_-NPs with sizes of 30–60 nm played positive roles; however, particles of 5 and 135 nm posed negative effects, indicating that Al_2_O_3_-NPs with various sizes affected the membrane permeability of mitochondria and the activity of the TCA cycle [99]. In addition, soybean treated with a mixture of Ag-NPs, nicotinic acid, and KNO_3_ displayed better growth than that treated with Ag-NPs alone, which was through regulation of activated protein quality control for misfolded proteins [45]. Collectively, these results indicate that NP treatment promotes soybean survival from flooding mainly through maintaining mitochondria function, which could alleviate the energy shortage and lipid oxidation induced by flooding stress.

### 4.3. Omic Analysis Using Flooding-Tolerant Materials

Plant omics of transcriptomics, proteomics, and metabolomics provides further insights into the inner workings of plant cells, cell–cell communications, and plant–environment interactions [57]. It was shown that genes related to protein folding and RNA modification were associated with the improvement of flooding tolerance, which was evidenced with transcriptomic analysis in Qihuang 34 [105] and Embrapa 45 [126], which are flood-tolerant soybean cultivars. Similar results were obtained from proteomic studies using flood-tolerant materials of mutant and ABA-treated seedlings, showing that protein synthesis and RNA regulation-related proteins triggered soybean tolerance to initial flooding [38]. Together with RNA regulation and protein metabolism, hormone response contributed to initial flooding tolerance through the inhibition of cytochrome P450 77A1 [127]. Moreover, metabolomic study of soybean showed that most altered compounds were enriched in carbon–nitrogen metabolism and phenylpropanoid pathway, and especially an increased level of sucrose was observed in flooding-sensitive cultivars than in tolerant cultivars [128]. Integrated omic data derived from proteomics and metabolomics indicated that fructose was a critical metabolite to assist in soybean flooding tolerance at initial stages via regulation of hexokinase and phosphofructokinase [51]. Taken together, proteomics, in combination with transcriptomics and metabolomics, improves the capability to uncover prospective flood tolerance responses in soybean as for RNA modification, protein synthesis, and sugar catabolism.

## 5. Differences between Response and Tolerant Mechanisms against Flooding Stress

Flooding response within stress and during the post-recovery stage as well as flood tolerance at the initial stage were characterized in soybean based on proteomics [20,22,23]. With the integration of plant omics data, which were collected from uniform experimental conditions, a schematic of flooding response and tolerance in soybean root is presented in Figure 1.

RNA metabolism plays a role in flooding tolerance, which was indicated by increased protein abundance and upregulated transcript levels of eukaryotic aspartyl protease and glycine-rich RNA-binding protein 3 in flood-tolerant materials compared with wild-type soybean [38]. Proteases participated in protein quality control, protein degradation, and cell lysis in response to environmental stimuli [129]. It was reported that overexpression of *AtASPG1*, an aspartic protease gene, enhanced drought tolerance and it functioned in ABA-dependent pathways in *Arabidopsis* [130]. Furthermore, AtASPG1 localized in the endoplasmic reticulum, indicating its possible roles in protein processing [130]. A glycine-rich RNA-binding protein 3 was cloned from cucumber fruit, namely *CsGR-RBP3*, and *Arabidopsis* overexpressed *CsGR-RBP3* displayed lower ROS levels and faster growth under chilling stress [131]. Moreover, CsGR-RBP3 localized in the mitochondria and its expression level was downregulated by ABA biosynthesis inhibitor [131]. These results indicate that eukaryotic aspartyl protease and glycine-rich RNA-binding protein 3 improve soybean flooding tolerance by regulating protein metabolism and ROS scavenging. In addition, it suggests that the endoplasmic reticulum and mitochondria might be susceptible to flooding.

Protein metabolism was shown to trigger flooding tolerance in soybean, and proteins including nascent polypeptide associated complex, chaperone proteins, as well as ATPase family AAA domain containing protein 1 were relevant to protein synthesis, folding, and degradation [38,42,45,127]. Nascent polypeptide associated complex accumulated in flood-tolerant materials compared with wild-type plants in response to initial flooding [38]. It interacted with polypeptides and worked as an integral component for protein folding, which occurred in the calnexin cycle with the help of an array of chaperone proteins [132]. In the calnexin cycle, chaperones of calnexin, calreticulin, and chaperone 20 increased in flood-tolerant materials [38,45]; however, inhibition of the calnexin cycle in flooded wild-type plants accelerated the accumulation of misfolded proteins, thereby promoting protein degradation. Degradation of misfolded proteins is critical to reestablish homeostasis equilibrium for protein metabolism [133]. It was reported that ubiquitin-mediated proteolysis was activated in flooded wild-type soybean [16], while the transcript level of ubiquitination-related protein ATPase family AAA domain containing protein 1 was downregulated in flood-tolerant materials [127]. Collectively, this indicates that enhanced polypeptide synthesis and protein folding lighten the load of misfolded proteins, ultimately ensuring the high efficiency of protein quality control under flooding.

A consequence of flooding stress is the requirement for energy conservation, which is invoked through adjustments in gene expression, carbohydrate catabolism, NAD(P)^+^ regeneration, and ATP production [107]. Imbalance of carbohydrate metabolism was associated with flooding injury in wild-type soybean [50], and protein abundance of phosphofructokinase [51], glyceraldehyde-3-phosphate dehydrogenase [53], as well as enolase [38] was further accelerated in flood-tolerant materials compared with wild type. Under flooding, down-regulated hexokinase and enhanced glycolysis were observed in flood-tolerant materials compared with wild type [23,51]. Besides this, glucose degradation and sucrose accumulation were accelerated in flooded wild-type soybean [50]; however, contents of sucrose and glucose were reduced in flood-tolerant materials [51]. Pyruvate is the end metabolite of glycolysis, and it enters fermentation and the TCA cycle for ATP generation. With regard to fermentation, increased alcohol dehydrogenase in flooded wild-type soybean was further accumulated in flood-mutant lines [44,50], and overexpression of *GmAdh2* conferred soybean flooding tolerance [93]. It was shown that metabolites involved in the TCA cycle that were induced by flooding were further accumulated in flood-tolerant materials [51,52]. Moreover, under flooding, ATP production derived from the TCA cycle was reduced [65]; however, the TCA cycle was enhanced in flood-tolerant materials compared with wild type [53]. Taken together, these results imply that soybean copes with energy crisis from flooding stress by relying on the activation of glycolysis, fermentation, and the citrate cycle.

Cell-wall architecture enables plants’ adaptation to environmental stimuli. Flooding suppressed cell-wall lignification in wild-type soybean; however, it was recovered in flood-tolerant materials [38]. Besides this, flooding induced alterations of proteins associated with cell-wall metabolism in soybean. For example, expansin-like B1-like proteins, which were considered as flooding injury-associated indicators, accumulated depending on flooding severity [17]; however, they did not change in flood-tolerant materials [38]. Expansins played roles in cell-wall loosening and overexpression of *NtEXP4* conferred tobacco tolerance to salt and drought stresses [134]. Polygalacturonase-inhibiting protein 1 was another cell-wall-related protein, and it dramatically accumulated in flood-tolerant materials compared with wild-type soybean under flooding [38]. It was reported that polygalacturonase-inhibiting proteins inhibited fugal polygalacturonases, which degraded cell-wall pectin and caused cell-wall degradation [135]. Moreover, up-regulated polygalacturonases played roles in the programmed cell death of root cortical cells in maize under waterlogging [136]. Therefore, it was indicated that accumulated polygalacturonase-inhibiting protein 1 might suppress cell-wall degradation and eliminate cell death. Under flooding, cell death was an extensive injury in soybean root [17]; however, it was less evident in the flood-tolerant line than wild-type soybean [44]. In addition, Yin et al. [127] reported that matrix metalloproteinase was responsible for cell death and it was down-regulated in flood-tolerant materials compared to wild-type soybean. Taken together, these results indicate that reinforcement of cell-wall plasticity and cell-wall thickening improve soybean adaption to flooding stress.

Current developments in soybean omics provide the capability to identify genes, proteins, metabolites, as well as metabolic pathways that are associated with flooding response and tolerance, which were discussed above. Integrated omic data pointed out the core set of metabolic pathways in flooded soybean, including RNA metabolism, protein metabolism, carbohydrate metabolism, cell wall metabolism, and cell death (Figure 2). Consistent with these findings in flooded soybean, it was suggested that under flooding, the most promising traits related to morphological adaptations included enhanced root porosity, a barrier against radial oxygen loss, and the formation of adventitious roots; however, metabolic adaptations were related to the antioxidant system and carbohydrate availability [137]. The whole genome sequence of soybean Williams 82 was available in 2010 [138], which facilitated the fine mapping of genes located on quantitative trait loci (QTLs) enriched chromosomes that were associated with flooding tolerance. Since different genetic backgrounds might demonstrate varying genetic mechanisms, a pan-genome of wild and cultivated soybeans was constructed to capture the entire genomic diversity, which helped to link genetic variations for genes responsible for agronomic traits [139]. Recently, the SUB1, a QTL for the submergence tolerance for rice, was demonstrated to play a role in the combination of submergence and long-term water stagnation [140]. Therefore, access to pan-genome, QTLs, and omic datasets is essential to facilitate the identification of flood-tolerant mechanisms in soybean.

## 6. Interaction of Molecular Mapping and Plant Omics to Flooding Tolerance in Soybean

QTLs associated with agronomic traits represent a reservoir of alleles for breeders to create improved varieties [141]. Flooding tolerance is a complex trait, which is controlled with multiple genes and affected by interactions within genotype and stress conditions. Acquired QTLs and plant omics associated with flooding tolerance in soybean are presented in Figure 3.

In soybean, dozens of QTLs associated with flooding tolerance were identified. For seed-flooding tolerance, four QTLs were detected based on examination of germination rate and normal seedling rate [142], and 33 QTLs were identified using relative seedling length as a flooding tolerance indicator [143]. At the early vegetative growth stage, seven [13] and 20 QTLs [144] associated with flooding tolerance were obtained through phenotypic analyses of chlorophyll content, dry weight of shoot, and seed weight. Notably, seven QTLs were mapped for hypoxia-tolerant index of root development traits [145]. Furthermore, a tolerant allele of *qWT_Gm03* promoted soybean growth under waterlogging stress through regulation of root architecture and plasticity [146]. Chromosome region analysis showed that these QTLs for flooding tolerance were mainly located on chromosomes 4, 9, 10, 12, 13, and 14 [144,145,146]. In addition, proteomic studies indicated that chromosomes 5, 10, 11, and 13 contained abundant flooding response genes [20,23]; however, chromosome 17 contained more tolerant genes [23]. As a consequence of plant omic studies in soybean, 16 candidates associated with flooding tolerance were collected from transcriptomics and proteomics [23,38,51,53,93,127]. Although QTL hotspots assist in fine mapping and positional clone for genes responsible for flooding tolerance, normally, several candidates are involved in a specific region of the chromosome. Plant omics provides a large dataset of genes and proteins with precise locations on the chromosome based on a well-annotated genome derived from high-throughput sequencing. Taken together, a complementary strategy of QTL mapping and plant omics assists in efficient screening for flood-tolerant genes in soybean.

## 7. Conclusions

Proteomics with subcellular and PTMs is a reasonable tool for the elucidation of flood response mechanisms in soybean. Under flooding, proteins related to signaling, stress, and the antioxidative system are increased in the plasma membrane; ROS scavenging enzymes are suppressed in the cell wall; protein translation is suppressed through inhibition of preribosome biogenesis- and mRNA processing-related proteins in the nucleus; proteins involved in the electron-transport chain are reduced in the mitochondrion; and protein-folding related proteins decrease in the endoplasmic reticulum. The importance of PTMs reveals the core flooding responses of hormone signaling, energy metabolism, protein folding, and degradation. Furthermore, stress tolerance might be indicated to proceed with this mechanism reversely. The development of flood-tolerant soybeans using mutants and varieties as well as transgenic lines is important to interpret flood-tolerant mechanisms. Understanding plant response/tolerance mechanisms will aid in formulating strategies aimed at improving stress tolerance in crops. Especially, access to QTLs and omic datasets is essential to facilitate the identification of flood-tolerant mechanisms in crops. The information in this review has led to the development of biotechnological tools for developing crops with altered flood tolerance, as well as the identification of marker proteins/genes for stress-tolerant crops. These proteomic technologies connected with agriculture might become a good example in the field of plant science.

## Figures and Tables

**Figure 1 ijms-21-07497-f001:**
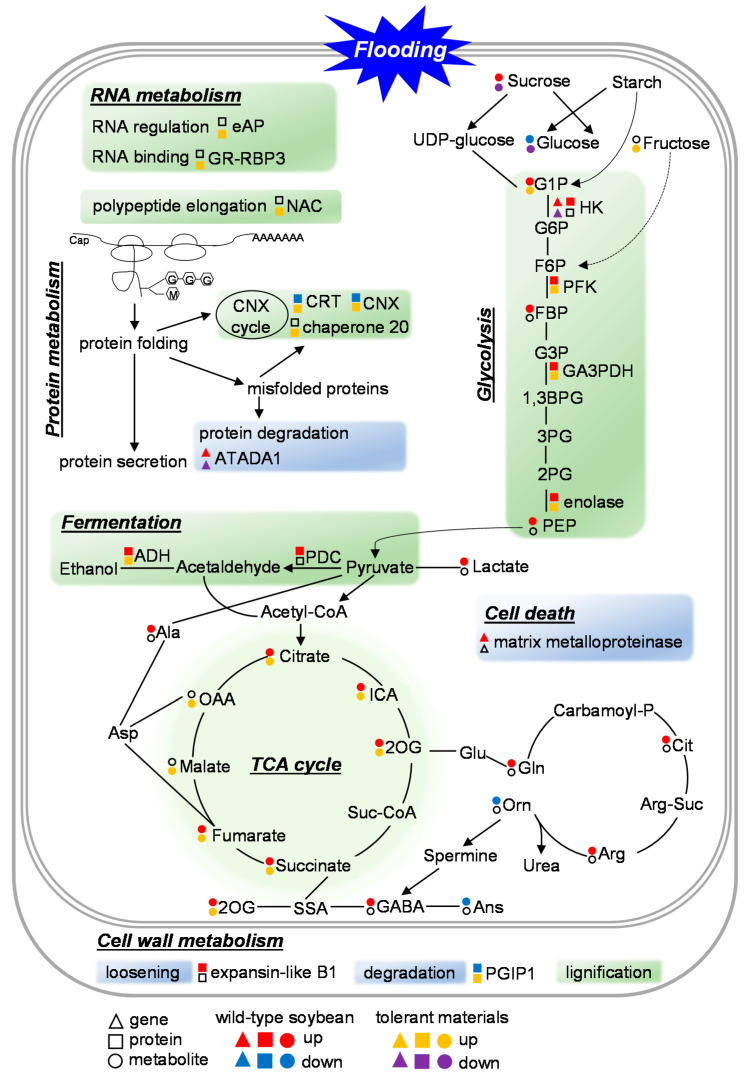
Differences between response and tolerant mechanisms against flooding stress. Scheme of core cellular pathways between response and tolerance in root of early-stage soybean in response to flooding stress is summarized based on plant omics data collected from wild-type soybean and flood-tolerant materials. Triangle, box, and circle indicate gene, protein, and metabolite, respectively. Red and blue colors indicate up- and down-regulation of gene/protein/metabolite in flooded wild-type soybean compared with untreated plant, respectively. Orange and purple colors indicate up- and down-regulation of gene/protein/metabolite in flood-tolerant materials compared with wild-type soybean under flooding, respectively. Gene, protein, and metabolite in black color mean the alteration induced by flooding stress is not clear. Green and blue colors indicate activation and suppression of metabolism in flood-tolerant materials compared with wild-type soybean under flooding. Abbreviations: 1,3BPG, glycerate 1,3-bisphosphate; 2OG, 2-oxoglutarate; 2PG, 2-phospho-glycerate; 3PG, 3-phospho-glycerate; ADH, alcohol dehydrogenate; Ala, alanine; Ans, anserine; Arg, arginine; Arg-Suc, arginino-succinate; Asp, aspartic acid; ATADA1, ATPase family AAA domain containing protein 1; Cit, citrulline; CNX, calnexin; CRT, calreticulin; eAP, eukaryotic aspartyl protease; FBP, fructose 1,6-bisphosphate; G, glucose; G1P, glucose 1-phosphate; G3P, glyceradehyde 3-phosphate; G6P, glucose 6-phosphate; GA3PDH, glyceraldehyde 3-phosphate dehydrogenase; GABA, gamma-aminobutyric acid; Gln, glutamine; Glu, glutamic acid; GR-RBP3, glycine-rich RNA-binding protein 3; HK, hexokinase; ICA, isocitrate; M, mannose; NAC, nascent polypeptide-associated complex; OAA, oxaloacetate; Orn, ornithine; PGIP1, polygalacturonase-inhibiting protein 1; PDC, pyruvate decarboxylase; PEP, phosphoenopyruvate; PFK, phosphofructokinase; SSA, succinate semialdehyde; Suc-CoA, succinyl-coenzyme A.

**Figure 2 ijms-21-07497-f002:**
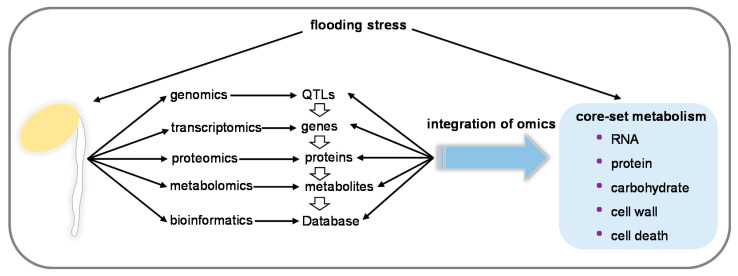
Overview of plant omics findings in response to flooding in soybean. Quantitative trait loci (QTLs), genes, proteins, and metabolites that are related to flooding stress were identified through genomics, transcriptomics, proteomics, and metabolomics, respectively. Soybean Proteome Database was constructed by bioinformatic analysis of omics data. Integration of soybean omics highlighted core set of flooding metabolisms in soybean.

**Figure 3 ijms-21-07497-f003:**
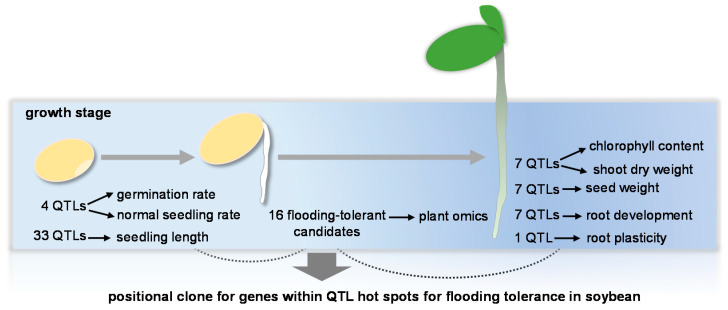
Interaction from QTL analysis and plant omics for flooding tolerance in soybean. QTLs associated with flooding tolerance in seed and at early vegetative growth stages are presented. Flood-tolerant indicators employed in phenotypic evaluation for QTL analysis are indicated. Flood-tolerant candidates were collected from transcriptomic and proteomic studies of soybean, including alcohol dehydrogenase 2, ATPase family AAA domain-containing protein 1, calnexin, calreticulin, chaperone 20, cytochrome P450 77A1-like, enolase, expansin-like proteins, eukaryotic aspartyl protease, glucose-6-phosphate isomerase 1, glyceraldehyde 3-phosphate dehydrogenase, glycine-rich RNA-binding protein 3, hexokinase, matrix metalloproteinase, nascent polypeptide-associated complex, and phosphofructokinase. Blue background indicates flooding stress.

**Table 1 ijms-21-07497-t001:** Utilization of proteomics in other crops under flooding stress.

Crop	Organ	Growth Stage/ Flooding Time	Findings	Reference
rice	coleoptile	seed/4 days	The majority of identified proteins were related to stress response and redox metabolism in anoxic rice coleoptile.	[76]
	spike	57-day-old/8 days	Electron transfer chain was destroyed to inhibit photosynthesis, while antioxidant system was activated to regulate ROS metabolism under submergence stress.	[77]
maize	leaf	4-leaf-stage/5 days	Proteins related to energy metabolism, photosynthesis, PCD, phytohormone, and polyamine responded to flooding; and damaged photosynthetic system led to disruption in energy metabolism and ROS overproduction under flooding.	[79]
	root	2-leaf-stage/3 days	NADP-malic enzyme, glutamate decarboxylase, coproporphyrinogen III oxidase, GSH S-transferase, GSH dehydrogenase, and XTH 6 were specifically accumulated to manage energy consumption, maintain pH levels, and minimize oxidative damage in waterlogging-tolerant maize.	[80]
	leaf	29-day-old/4, 28, 52 h	Combination of native IEF-PAGE and hrCNE was powerful to investigate alteration of Class III peroxidases, which played roles in ROS scavenging, cell-wall loosening, and aerenchyma formation in flooded maize.	[81]
	leaf	3-leaf-satge/6 days	6-BA exaggerated waterlogging defense through proteins related to protein metabolism, ROS scavenging, and fatty acid metabolism.	[82]
wheat	root	2-day-old/2 days	Decreased proteins of methionine synthase, beta-1,3-glucanases, and beta-glucosidase played roles in methionine assimilation and cell wall hydrolysis under flooding.	[83]
	leaf	7 days after anthesis/7 days	Waterlogging priming induced proteins related to energy metabolism, stress defense, and ethylene biosynthesis to improve wheat tolerance towards waterlogging.	[84]
	root	12-day-old/1–3 days	Acid phosphatase, oxidant protective enzyme, and SAM1 could be utilized as indicators for improving waterlogging tolerance in wheat.	[85]
	radicle	seed/1 day	Starch-sucrose metabolism was specifically enriched by submergences compared with salt and drought during seed germination.	[86]
rapeseed	root	1.5-day-old/4, 8, 12 h	Flooding induced proteins were mainly enriched in oxidation-reduction process, signal transduction, carbohydrate metabolism regardless of rapeseed genotype; however, large number of flood-altered-proteins indicated a quick active proteome response in the tolerant cultivar.	[87]
barley	root, leaf	4-leaf-stage/21 days	Proteins of PDC, ACO, and GST played roles in energy metabolism, ethylene production, and ROS homeostasis to improve waterlogging adaptation.	[88]
alfalfa	leaf	35-day-old/12 days	Amylase, ERF, CIPKs, GPX, and GST conferred alfalfa waterlogging tolerance.	[89]

6-BA, 6-benzyladenine; ACO, 1-amino cyclopropane 1-carboxylic acid oxidase; CIPK, calcineurin B-like interacting protein kinase; ERF, ethylene response factor; GPX, glutathione peroxidase; GS, glutamine synthetase; GSH, glutathione; GST, glutathione-S-transferase; hrCNE, high resolution Clear Native Electrophoresis; IEF, isoelectric focusing; PAGE, polyacrylamide gel electrophoresis; PCD, programmed cell death; PDC, pyruvate decarboxylase; ROS, reactive oxygen species; SAM1, S-adenosylmethionine synthetase 1; XTH, xyloglucan endotransglucosylase. Publications from 2010 onwards on crop proteomics under flooding stress were collected.

**Table 2 ijms-21-07497-t002:** Proteomic analysis to understand flood-tolerant mechanisms in soybean.

Experimental Materials	Growth Stage/ Flooding Time	Findings	Reference
radicle/128 soybean cultivars	2-day-old/2 days	Levels of RNA-metabolism related proteins and flooding indicator proteins correlated with flooding tolerance levels in soybean.	[31]
root/flooding mutant	2-day-old/2 days	Anaerobic metabolism was more efficient in mutant line than wild-type soybean under flooding, and reduction of cell-wall loosening allowed rapid growth of root tip after water removal.	[44]
root, hypocotyl/*GmAdh2*-overexpressed soybean	2-day-old/2 days	Overexpression of *GmAdh2* induced alternation of carbon flow with glycolysis and alcohol fermentation, improving germination under flooding.	[93]
root, hypocotyl/millimeter-wave treatment	2-day-old/2 days	Millimeter-wave irradiation promoted soybean recovery from flooding via regulation of glycolysis and redox-related pathways.	[94]
root/ABA treatment	2-day-old/2 days	ABA conferred soybean flooding tolerance through regulation of glycolysis and nuclear-localized proteins of zinc finger proteins, cell division cycle 5, and transducin.	[54]
root, hypocotyl/smoke treatment	2-day-old/2-day flood followed by 4-day recovery	Smoke enhanced soybean recovery from flooding via regulation of carbohydrate metabolism, glycolysis, and cell-wall components.	[95]
	2-day-old/2 days	Smoke promoted root growth of flooded soybean via energy production, ROS scavenging, activated ornithine synthesis, and suppressed ubiquitin proteasome.	[96,97]
root, leaf/Al_2_O_3_-, ZnO-, Ag-NPs treatment	7-day-old/3 days	Abundance of proteins involved in oxidation-reduction, stress signaling, and hormone pathway was principal for optimum growth of soybean under flooding in presence of Ag-NPs compared with Al_2_O_3_- and ZnO-NPs.	[98]
root, hypocotyl/Al_2_O_3_-NPs treatment	2-day-old/1, 2, 3, 4 days	Al_2_O_3_-NPs facilitated soybean acclimation to flooding via limited cell death, activated aerobic pathway, and ascorbate glutathione pathway.	[99,100]
	2-day-old/2-, 4-day flood followed by 2- and 4-day recovery	*S*-adenosyl-l-methionine-dependent methyltransferases and enolase were associated with flooding recovery in presence of Al_2_O_3_-NPs.	[101]
root, cotyledon/Ag-NPs treatment	2-day-old/2, 4 days	Under flooding, chemically synthesized Ag-NPs shifted fermentation to normal cellular process, while biosynthesized Ag-NPs enhanced protein degradation and ATP content.	[102,103,104]
	2-day-old/2 days	Mixture of Ag-NPs, nicotinic acid, and KNO_3_ exerted positive effect on soybean growth under flooding through regulation of protein quality control of misfolded proteins in the ER.	[45]

ABA, abscisic acid; ADH, alcohol dehydrogenase; NPs, nanoparticles; ROS, reactive oxygen species. Data have been collected from 2010 to 2020.

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
