# Peer review of "Review: Proteomic Techniques for the Development of Flood-Tolerant Soybean"

_ijms, 2020, doi:10.3390/ijms21207497_

Round 1

Reviewer 1 Report

This is a well written paper to give scientific information for flood tolerance in soybean specially focus on proteomics.

I thinks there is some missing part. It is necessary to discuss the relationship between QTLs and candidate genes, and proteomic data.

Author Response

This is a well written paper to give scientific information for flood tolerance in soybean specially focus on proteomics.

I think there is some missing part. It is necessary to discuss the relationship between QTLs and candidate genes, and proteomic data.

Answer: Thank you very much for your comment. As required, a new section 6 and figure 2 have been prepared to discuss the relationship of QTLs, candidate genes, and proteomic data that associated with flooding tolerance in soybean. These corrections have been added in new section entitled “6. Interaction of Molecular Mapping and Plant Omics to Flooding Tolerance in Soybean” in revised manuscript and marked with red color.

Reviewer 2 Report

The review, based on its title, aims to overview proteomic approaches to help breeding flooding tolerant soybean cultivars. The authors are experts of the field indicated by the fact that they write reviews about the same topic almost every year (Komatsu, S.; Tougou, M.; Nanjo, Y. Proteomic techniques and management of flooding tolerance in soybean. J. Proteome Res. 2015, 14, 3768-3778.;  Yin, X.; Komatsu, S. Comprehensive analysis of response and tolerant mechanisms in early-stage soybean at initial-flooding stress. J. Proteomics 2017, 169, 225-232.; Wang, X.; Komatsu, S. Proteomic approaches to uncover the flooding and drought stress response mechanisms in soybean. J. Proteomics 2018, 172, 201−215.). This, however, means that the reviewed/discussed papers/results largely overlap. In the present review, the authors aim to comprehensively overview all earlier work on soybean flooding tolerance and repeat many statements/conclusions they have made in their earlier reviews.  Moreover, they included three large Tables into their review to summarize all previous proteomic (or in generall „omic”) results reported for soybean flood response/tolerance and than discribe the same information in the text. Finally they describe and discuss it again in the  section5. Differences between Response and Tolerant Mechanisms against Flooding Stress.”. Altogether it means that most of the information is triplicated in the current review in addition to their appearance in previous reviews.  

It would be advisable to focus the review for recent additions, novel findings  to our previous knowledge on soybean flooding tolerance previously reviewed by the authors and to avoid repetitions within the current review. Thus, this riview could be much more concise and better focused. It would help the reader a lot since in the present form the review is very hard to read, which is also due to the very weak English of the manuscript. A manuscript file with some more detailed comments is uploaded.

Author Response

Reviewer 2

The review, based on its title, aims to overview proteomic approaches to help breeding flooding tolerant soybean cultivars. The authors are experts of the field indicated by the fact that they write reviews about the same topic almost every year (Komatsu, S.; Tougou, M.; Nanjo, Y. Proteomic techniques and management of flooding tolerance in soybean. J. Proteome Res. 201514, 3768-3778.; Yin, X.; Komatsu, S. Comprehensive analysis of response and tolerant mechanisms in early-stage soybean at initial-flooding stress. J. Proteomics 2017169, 225-232.; Wang, X.; Komatsu, S. Proteomic approaches to uncover the flooding and drought stress response mechanisms in soybean. J. Proteomics 2018, 172, 201−215.). This, however, means that the reviewed/discussed papers/results largely overlap. In the present review, the authors aim to comprehensively overview all earlier work on soybean flooding tolerance and repeat many statements/conclusions they have made in their earlier reviews.  

Moreover, they included three large Tables into their review to summarize all previous proteomic (or in general “omic”) results reported for soybean flood response/tolerance and then describe the same information in the text.

Answer: Thank you very much for your comments. Table size has been reduced, and descriptions between tables and text have been modified. furthermore, table contents have been corrected and marked with red color in revised manuscript.

Finally, they describe and discuss it again in the section “5. Differences between Response and Tolerant Mechanisms against Flooding Stress.”. Altogether it means that most of the information is triplicated in the current review in addition to their appearance in previous reviews.  

Answer: Thank you very much for your comments. The discussions in section 5 have been modified and focused on flooding-tolerant candidates. The corrections have been marked with red color in revised manuscript.

It would be advisable to focus the review for recent additions, novel findings to our previous knowledge on soybean flooding tolerance previously reviewed by the authors and to avoid repetitions within the current review. Thus, this review could be much more concise and better focused.

Answer: As suggested, repetitions in the current review have been carefully avoided. Furthermore, new session, which is “6. Interaction of Molecular Mapping and Plant Omics to Flooding Tolerance in Soybean”, has been prepared for discussion of current idea.

It would help the reader a lot since in the present form the review is very hard to read, which is also due to the very weak English of the manuscript.

Answer: As suggested, this article has been corrected by native speaker.

Addition comments in PDF from reviewer 2:

Answer: Thank you very much for your correction in the manuscript. As required, the commented parts have been corrected and they were marked with red color in revised manuscript.

Round 2

Reviewer 1 Report

Authors described properly for relationship among QTLs and flooding stress.

Author Response

Thank you very much for your review.

Reviewer 2 Report

Thanks for the authors to work hard on the manuscript to improve it. However, they could not solve the main problem. Still he review largely describe the same works in Table and text formats what is not required. Still I see the review as descriptive and not realy constructive. Some general sentences at the end of sections that what is not understood is not enough. Still I lack a clear general overview how the different findings come together. Moreover, the authors still repeat in the first sections what they have reviewed earlier. In my view, they should write a more focused review on new results, which somewhere from section 5 onward. Furthermore, an outlook for what is known in other plant species would be very useful.

Author Response

Thanks for the authors to work hard on the manuscript to improve it. However, they could not solve the main problem.

Answer: We are grateful that reviewer gives authors the chance to improve the contents of this review article. Based on comments from reviewer, authors have corrected this review article.

Still the review largely describes the same works in Table and text formats what is not required.

Answer: Previous Tables 1 and 2 have been shifted as Supplemental Tables 1 and 2 after modification. Furthermore, sections 2 and 3 have been shortened. Corrected sentences have been marked with green color.

Still I see the review as descriptive and not really constructive. Some general sentences at the end of sections that what is not understood is not enough.

Answer: Thank you very much for your comment. The general sentences at the end of sections have been re-written with green color.

Still I lack a clear general overview how the different findings come together.

Answer: A new Figure 2 has been prepared to indicate a general overview of how different findings come together. The description of new Figure 2 has been added in section 5 with green color. Furthermore, previous Figure 2 has been shifted as new Figure 3.

Moreover, the authors still repeat in the first sections what they have reviewed earlier.

Answer: As suggested, the first section “1. Introduction” has been re-written. Corrected sentences have been marked with green color.

In my view, they should write a more focused review on new results, which somewhere from section 5 onward.

Answer: Thank you very much for your comment. A new paragraph has been written in section 5 to describe the general overview of current findings as well as new results that related to soybean. The new paragraph has been marked with green color.

Furthermore, an outlook for what is known in other plant species would be very useful.

Answer: As suggested, new section “3.3. The State of Proteomic Analysis of Other Crops under Flooding Stress” has been added with new Table 1. Furthermore, “Introduction” has been changed with knowledge on rice. Additional sentences have been marked with green color.

Round 3

Reviewer 2 Report

The review was sufficiently revised and now better focuses on novel aspects of flodding tolerance mechanisms.